# Identifying Causal Changes Between Linear Structural Equation Models

**Vineet Malik**[1]      **Kevin Bello**[2,3]      **Asish Ghoshal**[4]      **Jean Honorio**[5]

[1]Computer Science Department, Purdue University, West Lafayette, Indiana, USA
[2]Machine Learning Department, Carnegie Mellon University, Pittsburgh, Pennsylvania, USA
[3]Booth School of Business, University of Chicago, Chicago, Illinois, USA
[4]Meta AI, Seattle, Washington, USA
[5]School of Computing and Information Systems, The University of Melbourne, Melbourne, Australia

## Abstract

Learning the structures of structural equation models (SEMs) as directed acyclic graphs (DAGs) from data is crucial for representing causal relationships in various scientific domains. Instead of estimating individual DAG structures, it is often preferable to directly estimate changes in causal relations between conditions, such as changes in genetic expression between healthy and diseased subjects. This work studies the problem of directly estimating the difference between two linear SEMs, i.e. *without estimating the individual DAG structures*, given two sets of samples drawn from the individual SEMs. We consider general classes of linear SEMs where the noise distributions are allowed to be Gaussian or non-Gaussian and have different noise variances across the variables in the individual SEMs. We rigorously characterize novel conditions related to the topological layering of the structural difference that lead to the *identifiability* of the difference DAG (DDAG). Moreover, we propose an *efficient* algorithm to identify the DDAG via sequential re-estimation of the difference of precision matrices. A surprising implication of our results is that causal changes can be identifiable even between *non-identifiable* models such as Gaussian SEMs with unequal noise variances. Synthetic experiments are presented to validate our theoretical results and to show the scalability of our method.

## 1 INTRODUCTION

Structural equation models (SEMs) are effective models to express causal relationships among variables in a system (Pearl 2009, Peters et al. 2017). However, both the parameters and the graphical structure representing causal relations, typically assumed to be a directed acyclic graph (DAG), are *unknown*. In various fields, including computational biology. (Sachs et al. 2005, Hu et al. 2018, Friedman et al. 2000), epidemiology (Robins et al. 2000), medicine (Plis et al. 2010, 2011), and econometrics (Imbens 2020, Hoover et al. 2009, Demiralp & Hoover 2003), developing methods to estimate the underlying DAG structure from available data is of utmost importance. This task is commonly known as causal discovery or structure learning, and numerous algorithms have been proposed for this purpose in the past few decades.

In this work, we assume causal sufficiency, which means that there are no unobserved confounders. However, even under this assumption, it is generally not possible to identify the underlying DAG structure, and the problem remains NP-complete in general (Chickering 1996, Chickering et al. 2004). Popular methods like PC (Spirtes et al. 2000) and GES (Chickering 2003) require an additional assumption known as faithfulness (Uhler et al. 2013) to consistently estimate the Markov equivalent class of the true DAG in large samples. However, these methods are not consistent in high-dimensional settings (Ghoshal & Honorio 2017*a*, 2018) unless there is an assumption of sparsity or small maximum-degree of the true DAG (Kalisch & Bühlman 2007, Nandy et al. 2018, Van de Geer & Bühlmann 2013). As a result, the presence of hub nodes, which are commonly observed in many networks (Barabási & Albert 1999, Barabási et al. 2011, Barabasi & Oltvai 2004), adds significant complexity to the problem of learning the DAG.

However, in many cases, the main objective is to identify changes in the causal mechanisms between two or more related SEMs, rather than to estimate the full underlying DAG structure of each SEM. For instance, in root cause analysis, an operator may be interested in identifying the sources that explain the differences between the working and failure states of a microservices system (Ikram et al. 2022, Paleyes et al. 2023, Li et al. 2022). Recent work by Assaad et al. (2023) propose an approach to estimate the difference in causal changes between normal and anomalous regimes based on a causal graph of the normal regime to

detect root causes. Our work complements such methods by providing a framework to directly estimate the differences between two SEMs without requiring the individual DAG structures. In the context of biological pathways, genes have the ability to control different groups of target genes based on the cellular environment or the presence of specific disease conditions (Hudson et al. 2009, Pimanda et al. 2007). Although the individual DAGs may be *dense*, the number of causal changes could be *sparse* (Schölkopf et al. 2021, Tanay et al. 2005, Perry et al. 2022). An additional practical scenario where our problem setting is applicable includes time-varying models, as discussed by Giannakis et al. (2018), where data samples are divided into possibly overlapping windows. For linear SEMs, Natali et al. (2021) describe the model as $X_t = B_t X_t + e_t$. Considering scenarios where the strength of causal relationships diminishes over time, such as the waning efficacy of a vaccine against disease resistance, demonstrates the relevance of our problem setting in practical environments.

In more detail, we focus on the problem of identifiability of *causal* structural changes given samples from two related linear SEMs. We consider linear SEMs where the noise variances at each individual SEM are allowed to vary, and, moreover, these noises can have arbitrary distributions with finite mean and second moment. Crucially, we do not impose additional structural assumptions such as sparsity, small maximum degree, or bounded tree width on the individual DAGs.

**Our contributions.** Our work introduces two key innovations to this problem. First, we prove that the difference DAG (causal changes) are identifiable for general linear SEMs, including non-identifiable models such as Gaussian SEMs with unequal noise variances. Second, motivated by our identifiability conditions, we propose an *efficient* algorithm that scales to thousands of variables. More specifically:

1. We present novel sufficient conditions (Assumptions 2 and 3) for identifiability of the difference DAG between two linear SEMs.

2. We develop a polynomial-time algorithm for directly estimating the DDAG between two linear SEMs (Algorithm 1) and show that our two conditions, Assumptions 2 and 3 are necessary for Algorithm 1 to identify the DDAG.

3. Since our algorithm is agnostic to the type of estimator for the difference of precision matrices, we leverage recent progress in this area and implement an efficient method that scales to thousands of nodes.

Proofs for all theoretical results are provided in the Supplementary Material.

## 2 RELATED WORK

**Learning individual DAGs.** One way to identify causal changes (albeit inefficient) would be to estimate individual DAGs for each environment and then to test for structural differences between the two DAGs. Some classical and recent methods for learning DAGs from a single dataset include: Constraint-based algorithms such as PC and FCI (Spirtes et al. 2000); in score-based methods, we have greedy approaches such as GES (Chickering et al. 2004), likelihood-based methods (Peters & Bühlmann 2014, Loh & Buhlmann 2014, Aragam & Zhou 2015, Aragam et al. 2019, Hoyer et al. 2008), and continuous-constrained learning (Zheng et al. 2018, Bello et al. 2022, Deng, Bello, Aragam & Ravikumar 2023, Deng, Bello, Ravikumar & Aragam 2023). Order-based methods (Teyssier & Koller 2005, Larranaga et al. 1996, Ghoshal & Honorio 2018, Rolland et al. 2022, Montagna et al. 2023), methods that test for asymmetries (Shimizu et al. 2006, Bühlmann et al. 2014), and hybrid methods (Nandy et al. 2018, Tsamardinos et al. 2006). Additionally, recent recursive algorithms have been developed for causal structure learning, such as the method by Mokhtarian et al. (2021). Finally, note that in order to use these methods, the individual DAGs must be identifiable, which is not the case for Gaussian SEMs with unequal noise variances (Pearl 2009). The identifiability of Gaussian noises with equal variances were proven in Peters & Bühlmann (2014) and Loh & Buhlmann (2014); while the identifiability of linear non-Gaussian models is given in Shimizu et al. (2006). In fact, a key implication of our results is that we can identify causal changes even when individual DAGs are unidentifiable.

**Differences in undirected graphs.** The problem of learning the difference between undirected graphs (or Markov random fields) has received much more attention than the directed case. For instance, Zhao et al. (2014), Liu et al. (2017), Yuan et al. (2017), Fazayeli & Banerjee (2016) develop algorithms for estimating the difference between Markov random fields and Ising models with finite sample guarantees. See Zhao et al. (2022), Varici et al. (2021) for recent developments in this direction. Another closely related problem is estimating invariances between causal structure across multiple environments (Peters et al. 2016). However, this is desirable when the *common structure* is expected to be sparse across environments, as opposed to our setting where the *difference* is expected to be sparse.

**Differences in directed graphs.** The problem of estimating the difference between DAGs has been previously studied by Wang et al. (2018), Varici et al. (2022), Chen et al. (2023), Yang et al. (2024). Under the same setting as ours, Wang et al. (2018) developed a PC-style algorithm (Spirtes et al. 2000), which they call *DCI*, for learning the difference between the two DAGs by testing for invariances between regression coefficients and noise variances between the two

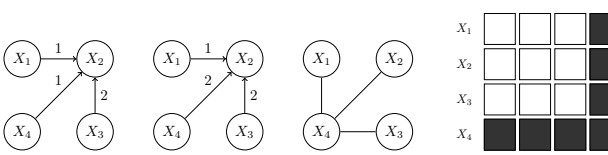

Figure 1: From left to right: the two SEMs, the difference undirected graph (or difference of moral graphs), and the difference of precision matrices between the two SEMs with non-zero entries shown in black.

models. However, sample complexity guarantees are hard to obtain for their method due to the use of many approximate asymptotic distributions of test statistics. Since the primary motivation behind directly estimating the difference between two DAGs is sample-efficiency, a lack of finite sample guarantees is a significant shortcoming. In contrast, our algorithm works by repeatedly eliminating vertices and re-estimating the difference of precision matrix over the remaining vertices. Thereby, we are able to leverage existing algorithms for computing the difference of precision matrix to obtain finite sample guarantees for our method. Furthermore, the DCI algorithm estimates regression coefficients (and noise variances) in the individual DAGs, while our method never estimates weights or noise variances of individual SEMs. Consider the example given in Figure 1 where the difference DAG contains only one edge $X_4 \rightarrow X_2$. In order to prune the edges $X_1 - X_4$ and $X_3 - X_4$ which are present in the difference undirected graph but not in the difference DAG, DCI would compute regression coefficients $\theta^1_{4|S}$ and $\theta^2_{4|S}$ for all $S \subseteq \{1, 2, 3\}$, where $\theta^1_{j|S}$ (resp. $\theta^2_{j|S}$) denotes regression coefficients obtained by regressing $X_j$ against $X_S$ in the first (resp. second) SEM. For linear SEMs, estimating regression coefficients is equivalent to estimating the precision matrix (Lemma 1 from Ghoshal & Honorio (2017b)). Furthermore, Danaher et al. (2014) have shown that directly estimating the difference between precision matrices is more sample efficient than estimating individual precision matrices and computing the difference.

## 3 PRELIMINARIES

We use $[p]$ to denote the set of integers $\{1 \ldots p\}$. For a matrix $A$, we will denote its $i$-th row (resp. $i$-th column) by $A_{i,*}$ (resp. $A_{*,i}$). Furthermore, we define the support of the matrix $A$, denoted as $\text{supp}(A)$, as the set of indices $(i, j)$ for which the entries of $A$ are non-zero, i.e., $\text{supp}(A) = \{(i, j) \mid A_{i,j} \neq 0, \text{ for } i, j \in [p]\}$.

Let $X = (X_1, \ldots, X_p)$ be a $p$-dimensional random vector. We will denote a structural equation model (SEM) by the tuple $(B, D)$ where $B$ is an autoregression matrix and $D = \mathbf{Diag}(\{\sigma_i^2\})$ is a diagonal matrix of noise variances. Then, the SEM $(B, D)$ defines the following generative model

over $X$:

$$X_i = B_{i,*}X + \varepsilon_i, \quad \forall i \in [p],$$

where the noises are mutually independent with $\mathbb{E}\left[\varepsilon_i\right] = 0$ and $\text{Var}\left[\varepsilon_i\right] = D_{i,i} = \sigma_i^2 < \infty$. In this work, the autoregression matrix $B$ encodes a directed acyclic graph (DAG) $G = ([p], \text{supp}(B))$ over $[p]$, where the edge $(i, j)$ denotes the directed edge $i \leftarrow j$.

**Remark 1.** *It is worth noting that distributions with bounded second moment include a large set of distributions such as Gaussian, uniform, Gumbel, exponential, Laplace, etc. This does not include distributions like the Cauchy distribution, which have infinite variance. Our class of SEMs covers a significant part of the classical LiNGAM models while also allowing for Gaussian distributions. Therefore, the classical linear non-Gaussian acyclic model (LiNGAM) (Shimizu et al. 2006) is a special case of our class of SEMs, with the added restriction of bounded variance.*

Given two SEMs, $(B^{(1)}, D)$ and $(B^{(2)}, D)$, our goal is to recover the structure of the difference between the two DAGs, that is, $\text{supp}(B^{(1)} - B^{(2)})$. Going forward, we use $\Delta_B$ to denote $B^{(1)} - B^{(2)}$ and $\Delta$ to denote the difference graph, $([p], supp(\Delta_B))$. We assume that the two DAGs, $G^{(1)}$ and $G^{(2)}$, share a topological ordering, thereby resulting in no edge reversals between them, which is a reasonable assumption in several practical problems (Zhao et al. 2014, Belyaeva et al. 2021). Formally, we are interested in the following problem:

**Problem 1.** *Given two sets of observations $X^{(1)} \in \mathbb{R}^{n_1 \times p}$ and $X^{(2)} \in \mathbb{R}^{n_2 \times p}$, drawn from the* unknown *SEMs $(B^{(1)}, D)$ and $(B^{(2)}, D)$ respectively, estimate $\Delta$.*

We will often index the two SEMs by $\kappa \in \{1, 2\}$. We will denote the difference between the precision matrices[1] of the two SEMs by: $\Delta_\Omega$, and the precision matrix over any subset of variables $S \subseteq [p]$ by $\Delta_\Omega^S$. Similarly, $\Omega^{(\kappa, S)}$ denotes the precision matrix over the subset $S$ in the SEM indexed by $\kappa$. We will denote the set of topological orderings induced by a DAG $G = ([p], E)$ by $\mathcal{T}(G) = \{(\tau_1, \ldots, \tau_p) \in \Pi([p]) \mid \forall i, j \in [p] \text{ if } i > j, (\tau_j, \tau_i) \notin E\}$, where $\Pi([p])$ is the set of permutations of $[p]$. The notation $i \preceq_\tau j$ denotes that the vertex $i$ comes before $j$ (or $i = j$) in the topological order $\tau$. For any $\tau \in \mathcal{T}(G)$, we will consider sequence of graphs $G_{[m,\tau]} = (V_{[m,\tau]}, E_{[m,\tau]})$, indexed by $(m, \tau)$, where $G_{[m,\tau]}$ is the induced subgraph of $G$ over the first $m$ vertices in the topological ordering $\tau$, i.e., $V_{[m,\tau]} = \{\tau_i \mid i \leq m\}$ and $E_{[m,\tau]} = \{(i, j) \in E \mid i, j \in V_{[m,\tau]}\}$. We use the term "terminal vertices" to denote the vertices in a DAG that have no outgoing edges.

---

[1]We use the standard definition of a precision matrix, i.e., the inverse covariance matrix.

Finally, we will always index precision matrices by vertex labels, i.e., $\Omega_{i,j}$ denotes the precision matrix entry corresponding to the $i$-th and $j$-th node of the graph.

# 4   MAIN RESULTS: NOVEL IDENTIFIABILITY CONDITION AND POLY-TIME ALGORITHM

In our analysis, we discuss different strategies for removing terminal vertices and their implications on the edge requirements for the difference graph $\Delta$. We explore two extreme cases: (i) removing terminals one-by-one, and (ii) removing terminals all-at-once. We establish the significance of the minimal topological layering of the difference DAG in this context and provide an example of how the edge requirements can be relaxed by considering the all-at-once removal strategy. Finally we establish the identifiability conditions of difference DAGs and present Algorithm 1, a poly-time algorithm to identify those DAGs.

## 4.1   TERMINAL VERTICES

Let $(B^{(1)}, D)$, $(B^{(2)}, D)$ be two Structural Equation Models (SEMs) such that they share at least one topological ordering. The difference precision matrix is defined as

$$\Delta_\Omega = \Omega^{(1)} - \Omega^{(2)}.$$

Using Proposition 2 from Ghoshal & Honorio (2018), the diagonal entries of difference precision matrix are given as

$$\Delta_{\Omega_{i,i}} = \sum_{l \in [p]} \frac{(B^{(1)}_{l,i} + B^{(2)}_{l,i})(B^{(1)}_{l,i} - B^{(2)}_{l,i})}{\sigma_l^2}. \quad (1)$$

From Equation 1, we derive the following proposition:

**Proposition 1.** *For any $i \in [p]$, if $i$ is a terminal vertex in $\Delta$, then $\Delta_{\Omega_{i,i}} = 0$.*

Recall that the two SEMs share a topological ordering, which is equivalent to saying that union of their DAGs is also a DAG, i.e. $G^\cup = G^{(1)} \cup G^{(2)}$ is a DAG. The terminals of $G^\cup$ is given by the intersection of set of terminals of DAGs of the two SEMs. Since $\Delta$ is a subgraph of $G^\cup$, we have the following proposition:

**Proposition 2.** *For any $i \in [p]$, if $i$ is a terminal vertex in $G^\cup$, then $i$ is a terminal vertex in $\Delta$.*

It is important to note that the converse of both Proposition 1 and Proposition 2 is not true in general.

The validity of converse of Proposition 1 is contingent upon certain weight conditions. In contrast, the converse of Proposition 2 hinges on structural constraints of the difference

graph $\Delta$. For the converse of Proposition 2 to hold, for every non-terminal vertex in $G^\cup$, at least one of their outgoing edge should also be in $\Delta$. So $\Delta$ must have at least $p - t$ edges, where $t$ is the number of terminals vertices. Hence, this proposition's converse does not universally hold but requires specific structural alignment within the graph. Finally the combined assumption for converse of Proposition 1 and Proposition 2 to hold, can be stated as follows:

**Assumption 1.** *For any $i \in [p]$, if $\Delta_{\Omega_{i,i}} = 0$, then $i$ is a terminal vertex in $G^\cup$.*

To find the incoming edges for these terminal vertices, we can examine the difference precision matrix $\Delta_\Omega$. By looking at the non-zero entries in the corresponding row or column of $\Delta_\Omega$ for each terminal vertex, we can identify the incoming edges for these vertices in the graph $\Delta$. This is given by the Lemma 1.

**Lemma 1.** *Under Assumption 1, for any $i \in [p]$, if $\Delta_{\Omega_{i,i}} = 0$, then $\forall j \in [p], \Delta_{\Omega_{i,j}} = -\frac{\Delta_{B_{i,j}}}{\sigma_i^2}$.*

This lemma allows us to identify the terminal vertices of $\Delta$ and the incoming edges to those terminals through the difference precision matrix. By iteratively removing these terminal vertices and repeating the process, we can recover the entire structure of $\Delta$, under the condition that Assumption 1 holds for the linear SEMs obtained from removing the terminal vertices. There are multiple ways to remove these terminal vertices, and in the subsequent sub-sections, we provide the analysis for two extreme cases: removing terminals one-by-one and removing them all-at-once. Assumption 1 is essentially the unification of the converses of both Proposition 1 and Proposition 2. Each of these propositions imposes distinct types of constraints on the SEMs. First, we will elucidate the structural implications of the converse of Proposition 2, particularly highlighting the significance of the minimal topological layering of $\Delta$. Subsequently, we will integrate this with the converse of Proposition 1 in Section 4.4.

## 4.2   ON THE FAILURE OF REMOVING TERMINALS ONE-BY-ONE

In this approach, we sequentially remove a single terminal vertex and its incoming edges from the difference graph $\Delta$ and both the SEMs. At each step, we re-estimate the difference precision matrix and re-apply Lemma 1 to identify the next terminal vertex and its incoming edges. This process is repeated until all vertices have been removed, and the entire structure of $\Delta$ is recovered. For this we need the converse of Proposition 2 to hold after removal of every terminal vertex.

Removing terminal vertices one-by-one is essentially removing vertices in reverse topological ordering. We can

formally state this as follows: for all topological ordering $\tau$ of $\Delta$, for all $m \in [p]$, $\tau_m$ is a terminal in $G^{(1)}_{[m,\tau]}$ and $G^{(2)}_{[m,\tau]}$.

This is equivalent to every topological ordering of $\Delta$ being also a valid topological ordering of $G^\cup$. Since $\Delta$ is a subgraph of $G^\cup$, every topological ordering of $G^\cup$ is a topological ordering of $\Delta$. However, the converse is not true in general. This highlights the importance of understanding the relationship between the topological orderings of $\Delta$ and $G^\cup$ when analyzing the structure of the difference graph. We further examine the conditions under which this is valid to gain further insight into the structural constraints necessary for the recovery of the entire structure of $\Delta$ through the iterative one-by-one removal of terminal vertices.

We introduce the concept of transitive edges in a DAG and explore their impact on topological orderings and the structure of the difference graph $\Delta$.

**Definition 1** (Transitive Edge[2]). *Let $G$ be a DAG. An edge $u \to v$ is called a transitive edge of $G$ if there exist multiple directed paths from $u$ to $v$ in $G$.*

Removing an edge from a DAG cannot decrease the set of topological orderings compatible with it; it can either remain the same or increase. The key property here is that it remains the same if and only if the removed edge is a transitive edge of the DAG.

**Proposition 3.** *The converse of Proposition 2 holds while removing terminals one-by-one if and only if all the edges of $G^\cup$ missing from $\Delta$ are transitive edges of $G^\cup$.*

Unfortunately, in the worst case, this can require $\Delta$ to be dense. For instance, consider $G^\cup$ to be the complete bipartite graph $K_{n,n}$ with the direction of all edges from one partition of $n$ vertices to the other partition of $n$ vertices. This graph has no transitive edges, i.e., all $n^2$ edges are non-transitive. Therefore, to identify $\Delta$ through the one-by-one removal process of the terminals, $\Delta$ must contain all the $n^2$ edges. In the next section, we present that removing all terminals at once will reduce this requirement from $n^2$ edges to just $n$ edges.

### 4.3 REMOVING TERMINALS ALL-AT-ONCE

In this alternative approach, we simultaneously remove all terminal vertices and their incoming edges from the difference graph $\Delta$ in one step. The difference precision matrix is then re-estimated and Lemma 1 is applied to identify all new terminal vertices and their incoming edges. This process is repeated until all the vertices are removed and the entire structure of $\Delta$ is recovered.

The converse of Proposition 2 states that the set of terminals of $\Delta$ is the same as the set of terminals $G^\cup$. This should hold even after all these common terminals are removed simultaneously. We first introduce the concept of topological layering of a DAG, which is a generalization of the well-known concept of topological ordering. Subsequently, we establish that the iterative necessity of the converse of Proposition 2 while removing terminals all-at-once is equivalent to minimal topological layering of $\Delta$ being a valid topological layering of $G^\cup$.

**Definition 2** (Topological Layering). *Let $G(V, E)$ be a DAG. A topological layering of $G$ is a partitioning of the vertex set $V$ into a sequence of sets $(L_0, L_1, ..., L_r)$, such that if $(u, v) \in E$, $u \in L_i$, and $v \in L_j$, then $i > j$.*

Each set of the partition corresponds to a layer. Essentially, a topological layering of G is a function $L : V \mapsto \{0, 1, 2, ..., r\}$, where $r < p$ and such that for every edge $(u, v) \in E, L(u) > L(v)$, that is, edges are allowed only to go from the vertices of a higher layer to the vertices of a lower layer. This concept generalizes topological ordering, since a topological ordering can be considered a special case where $r = p - 1$ and the function operates as a bijection. We then introduce the notion of minimal topological layering in which every vertex $v$ is assigned to the lowest possible layer $L(v)$ such that the condition of the topological layering still holds.

**Definition 3** (Minimal Topological Layering). *Let $G(V, E)$ be a DAG. The minimal topological layering of $G$ is a topological layering $L$ of $G$ such that there does not exist a topological layering $L'$ of $G$ with $L'(v) < L(v)$ for some $v \in V$.*

Note that the minimal topological layering of a DAG is unique. This is the topological layering that one obtains by recursively removing terminals of a DAG all-at-once as layers. Similarly, one can obtain a layering by recursively removing the roots (vertices with no incoming edges) of a DAG in an all-at-once fashion. This layering obtained by removing roots is used in many recent works on causal structure learning (Gao et al. 2020, Zhou et al. 2022, Park 2023). We observe that this layering of roots is the maximal topological layering of a DAG, where each vertex is assigned to the highest possible layer while maintaining the constraints of a topological layering. We now establish the link between the converse of Proposition 2 and the topological layering of $\Delta$.

**Lemma 2.** *The converse of Proposition 2 holds while removing terminals all-at-once if and only if the minimal topological layering of $\Delta$ is a valid topological layering of $G^\cup$.*

Therefore, the iterative requirement of the converse of Proposition 2 can be stated as the following assumption:

---

[2]Transitive edges have also been employed in (Bello & Honorio 2018) for learning DAGs.

**Assumption 2.** *The minimal topological layering of $\Delta$ is a valid topological layering of $G^{\cup}$.*

Assumption 2 is a significantly less stringent condition compared to requiring all topological orderings of $\Delta$ being topological orderings of $G^{\cup}$, which is equivalent to needing all topological layerings of $\Delta$ being topological layerings of $G^{\cup}$, Assumption 2 only asks for only one special topological layering of $\Delta$ to be compatible with $G^{\cup}$. Hence, under Assumption 2 $\Delta$ may have topological orderings that are not compatible with $G^{\cup}$.

From the unique minimal topological layering of a DAG, we define the notion of topological level of a vertex in a DAG. This concept of topological level will play a pivotal role in the following section.

**Definition 4** (Topological Level). *Let $G$ be a DAG and $L$ be its unique minimal topological layering. The topological level of a vertex $v$ in $G$ is defined as the value $L(v)$ assigned to that vertex by $L$.*

Going back to the example of the complete bipartite graph $K_{n,n}$ with the direction of all edges from one partition of $n$ vertices to the other partition of $n$ vertices, the topological level of all non-terminal vertices is one. For the minimal topological layering of $\Delta$ to be compatible with $G^{\cup}$, at least one edge from each non-terminal vertex of $G^{\cup}$ should be in $\Delta$. Hence, we only require $n$ of these edges to be part of $\Delta$. This is a significant relaxation compared to the $n^2$ requirement in case of removing terminals one-by-one.

In fact, for every DAG $G$, the minimal subgraph (in terms of the number of edges), in which topological levels of all vertices remain the same as in $G$, has $p - t$ edges, where $p$ is the number of vertices and $t$ is the number of terminal vertices, as for every non-terminal vertex, at least one of their outgoing edges to the immediate lower topological level should be part of the minimal subgraph. It is important to note that this minimal subgraph is not unique, and, in fact, there can be exponentially many such minimal subgraphs for a given DAG.

## 4.4 IDENTIFIABILITY OF DIFFERENCE DAGS

In this section, we investigate the intricacies of the converse of Proposition 1, combining it with Assumption 2 to produce the final identifiability condition.

Proposition 1 states that if $i$ is a terminal in $\Delta$, then $\Delta_{\Omega_{i,i}}$ is 0. The converse being true necessitates that $\Delta_{\Omega_{i,i}} = 0$ only if $i$ is a terminal in $\Delta$. This condition should hold level-wise as the condition in Assumption 2. First we introduce the concept of Diagonal Null levels of the difference precision matrix $\Delta_{\Omega}$, then we establish the relationship to the levels of the DAG structure. The diagonal entries $\Delta_{\Omega_{i,i}}$ of the difference precision matrix represent the difference in the

variances of the variable $i$ in the system. We define the concept of Diagonal Null (DN) levels as follows:

1. DN Level-0 variables: These are the variables that correspond to the indices $i$ where $\Delta_{\Omega_{i,i}} = 0$ in the original $\Delta_{\Omega}$, i.e., the diagonal entry of the difference precision matrix is zero.

2. DN Level-$k$ variables: For $k > 0$, these are the variables that become level-0 only after eliminating all DN level-0, DN level-1, ..., DN level-$(k-1)$ variables from the system and recalculating the difference precision matrix.

Hence, the iterative constraint of the converse of Proposition 1 becomes the following:

**Assumption 3.** *For every vertex, its DN level in $\Delta_{\Omega}$ is greater than or equal to its topological level in $\Delta$.*

Next, we demonstrate the necessity of the identifiability assumptions, Assumption 2 and Assumption 3. In other words, if either Assumption 2 or Assumption 3 is violated, it is possible to find an exponential number of pairs of SEMs that exhibit distinct DDAG structures while inducing the same difference precision matrix.

**Theorem 1.** *There exists an exponentially large set, $S(\alpha, \beta, \sigma)$, of pairs of SEMs, parameterized by $\alpha, \beta, \sigma \in \mathbb{R}_+$, s.t. for every pair of SEMs, $(B^{(1)}(\alpha, \beta), D(\sigma))$ and $(B^{(2)}(\alpha, \beta), D(\sigma))$, in $S(\alpha, \beta, \sigma)$, the identifiability Assumption 2 does not hold and produces the same difference precision matrix but distinct difference DAG.*

**Theorem 2.** *There exists an exponentially large set, $S(\alpha, \sigma)$, of pairs of SEMs, parameterized by $\alpha, \sigma \in \mathbb{R}_+$, s.t. for every pair of SEMs, $(B^{(1)}(\alpha), D(\sigma))$ and $(B^{(2)}(\alpha), D(\sigma))$, in $S(\alpha, \sigma)$, the identifiability Assumption 3 does not hold and produces the same difference precision matrix but distinct difference DAG.*

We note that Assumption 3 is not only theoretically significant but also holds in many practical scenarios. For instance, consider a situation where SEM 2 represents the interventional distribution of SEM 1, obtained through hard node interventions on SEM 1. In such cases, Assumption 3 is naturally satisfied. Similarly, this assumption is valid in scenarios where an agent external to the system performs stochastic do-interventions on any collection of variables. These instances are common in experimental designs and causal inference studies.

## 4.5 POLY-TIME ALGORITHM

With the above results in place, we are now ready to state our algorithm for directly learning the difference DAG.

We prove the correctness of Algorithm 1 in the population setting, i.e., when $\Sigma^{(\kappa)}$ is the true covariance matrix

**Algorithm 1:** Learning causal changes in linear SEMs

**Input:** $\Sigma^{(1)}$ and $\Sigma^{(2)}$

**Result:** $\Delta$

1  $V \leftarrow [p]$;
2  **while** $|V| > 1$ **do**
3     Estimate $\Delta_\Omega$ over $V$;
4     $S \leftarrow \{i \mid (\Delta_\Omega)_{i,i} = 0\}$;
5     **for** $i \in S$ **do**
6         $N_i \leftarrow \{j \mid (\Delta_\Omega)_{i,j} \neq 0\}$;
7         **for** $j \in N_i$ **do**
8             **if** $(j, i) \notin \Delta$ *and* $j \notin S$ **then**
9                 Add $(i, j)$ in $\Delta$;
10             **end**
11         **end**
12     **end**
13     $V \leftarrow V - S$;
14  **end**
15  **return** $\Delta$;

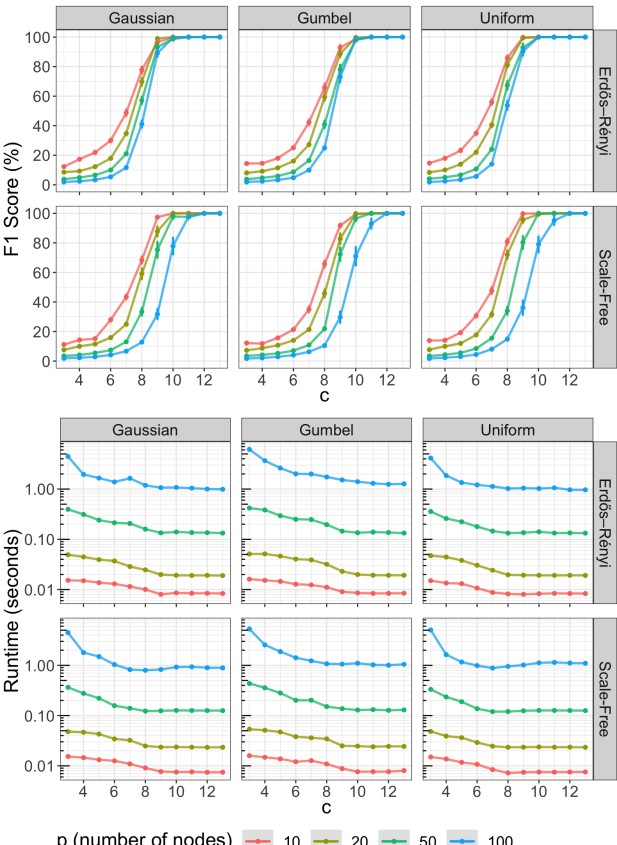

Figure 2: Performance vs sample size in low dimensions (up to 100 nodes). See Section 5.1 for details on the data generation process. We note that our method is capable of learning the DDAG in seconds and perfectly recovering the causal changes as the number of samples increases.

of the SEM $(B^{(\kappa)}, D^{(\kappa)})$ for $\kappa \in \{1, 2\}$. In this case $\Delta_\Omega$ can be computed efficiently by solving the linear system: $\Sigma^{(1)}(\Delta_\Omega)\Sigma^{(2)} = \Sigma^{(2)} - \Sigma^{(1)}$ (Zhao et al. 2014). Since $\Sigma^{(\kappa)}$ is positive definite, the above system has a unique solution.

**Theorem 3.** *Let $(B^{(1)}, D)$ and $(B^{(2)}, D)$ be two SEMs. Let $\Delta_B = B^{(1)} - B^{(2)}$ denote the difference between the two SEMs. Given the true covariance matrices $\Sigma^{(1)}$ and $\Sigma^{(2)}$, under Assumption 2 and Assumption 3, Algorithm 1 returns $\Delta$ such that $\Delta = ([p], \mathrm{supp}(\Delta_B))$.*

**Remark 2.** *It is known that Gaussian SEMs with unequal noise variances are known to be unidentifiable (Pearl 2009). Here we emphasize the surprising fact that we can identify causal changes even in non-identifiable models such as Gaussian SEMs with unequal noise variances.*

**Remark 3** (Computational Complexity)**.** *The computational complexity of Algorithm 1 is primarily influenced by the repeated estimation of the difference precision matrix, denoted as $\Delta_\Omega$, over the set $V$. Each estimation step is considered as an oracle call with a time complexity of $T(m)$, where $m$ represents the current size of the set $V$. Initially, the set $V$ contains $p$ elements, and in each iteration of the while loop, at least one vertex is removed from $V$, ensuring a maximum of $p$ iterations. Within each iteration, the for-loops over the sets $S$ and $N_i$ contribute an additional factor to the complexity, but this factor is bounded by $O(p^2)$ since it involves checking pairwise relationships in the worst case. Therefore, the overall computational complexity of the algorithm is $O(p \cdot T(m) + p^2)$. We invite the reader to see Figures 2, 3, and 4 for empirical runtimes of our algorithm.*

### 4.6 ON DIFFERENCE OF PRECISION MATRICES

The performance of our method depends on the accuracy with which the difference between the precision matrices is estimated. The problem of directly estimating the difference between the precision matrices of two Gaussian SEMs (or more generally Markov random fields), given samples drawn from the two individual models, has received significant attention over the past few years (Zhao et al. 2014, Belilovsky et al. 2016, Yuan et al. 2017, Liu et al. 2017, Jiang et al. 2018). Among these, the ADMM method of Jiang et al. (2018), the KLIEP algorithm of Liu et al. (2017), and the algorithm of Zhao et al. (2014) come with provable finite sample guarantees. We use the algorithm of Jiang et al. (2018) in our study, which is particularly effective when dealing with the sparse differences between two linear SEMs. This sparsity is also reflected in the differences between their precision matrices. For the two SEMs, the sparsity is evident in their precision matrix difference, denoted as $\Delta_\Omega = \Omega^{(1)} - \Omega^{(2)}$. As in (Jiang et al. 2018), we initially calculate the sample covariance matrices, $\hat{\Sigma}^{(1)}$ and

$\hat{\Sigma}^{(2)}$. Subsequently, a convex optimization problem is solved using ADMM, formulated as:

$$\hat{\Delta}_\Omega = \arg\min_{\Delta_\Omega} \left\{ \frac{1}{2} \text{Tr}\left( \Delta_\Omega^\top \hat{\Sigma}^{(1)} \Delta_\Omega \hat{\Sigma}^{(2)} \right) \right.$$
$$\left. + \text{Tr}\left( \Delta_\Omega (\hat{\Sigma}^{(1)} - \hat{\Sigma}^{(2)}) \right) + \lambda \|\Delta_\Omega\|_1 \right\},$$

where $\lambda$ is a regularization parameter. The approach of Jiang et al. (2018) is preferred in our work due to its computational efficiency, offering a complexity of $O(p^3)$. We also tested with other estimators in the literature such as Zhao et al. (2014) and the results were very similar, but much slower to obtain. We emphasize that our main contributions are related to the identifiability of the DDAG, and not to propose a new estimator of the difference of precision matrices. For our theory, the $\Delta_\Omega$ estimator is a black box. Thus, by using any estimator with guarantees, we implicitly borrow its conditions for correctness. The sample complexity of our algorithm follows straightforwardly from the sample complexity of the $\Delta_\Omega$ estimator; since we use the estimator by Jiang et al. (2018), we can make use of their finite-sample rates, see, for instance, Theorem 1 in Jiang et al. (2018).

In Figure 2, we explore the performance of Algorithm 1 by using the estimator of Jiang et al. (2018).

# 5 EXPERIMENTS

In this section, we describe the empirical results from the execution of our Alg. 1 on finite samples with the goal of verifying our theoretical results and showing the efficiency of our method on graphs of size up to $p = 1000$ nodes.

## 5.1 SYNTHETIC DATA GENERATION

For the generation of random SEM pairs, our approach starts with the construction of two random DAGs over $p$ nodes, adhering to the structural identifiability criteria outlined in Assumption 2. These random DAGs are designed to be individually dense, with an expected edge count of $O(p^{1.75})$, while ensuring that the difference DAG remains sparse, featuring an expected $O(p)$ edges. Our evaluation encompasses two prevalent models of random graphs: Erdős–Rényi graphs and Scale-Free graphs, where the latter are likely to generate hub-nodes, a known challenge for causal structure learning (Kalisch & Bühlman 2007). Importantly, our algorithm does not presuppose specific exogenous noise distributions. To this end, we assess performance across Gaussian, Gumbel, and Uniform noise distributions, with noise variance values selected uniformly at random from the interval $[0.25, 0.5]$. Additionally, edge weights are selected randomly from the combined intervals $[-0.25, -0.5] \cup [0.25, 0.5]$. If these sampled edge weights fail to satisfy Assumption 3, then we simply sample them again. After generating the pair of SEMs,

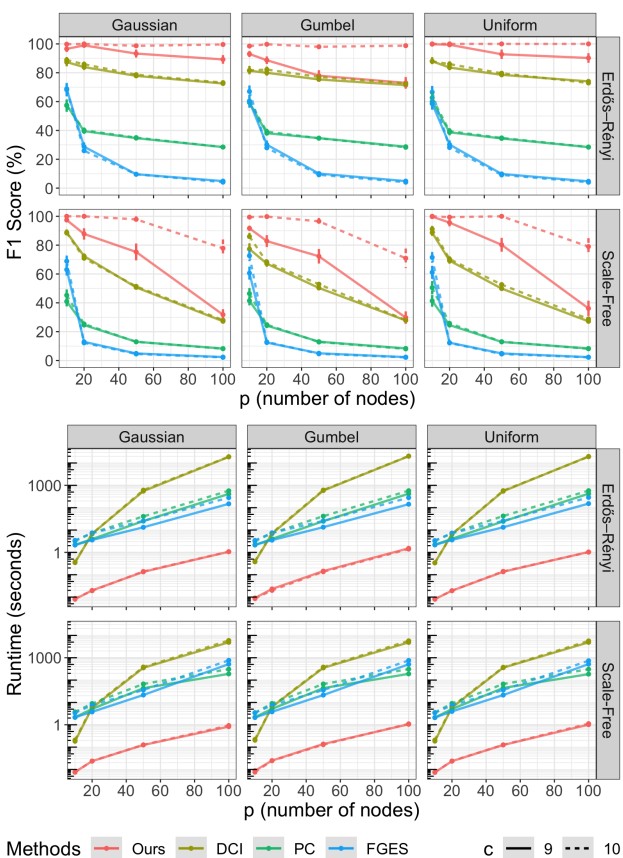

Figure 3: Performance vs number of variables (up to 100 nodes). See Section 5.1 for details on the data generation process. We note that our method outperforms direct learning methods such as DCI and indirect methods such as PC and GES in both recovery and time execution.

we generate $\lfloor e^c \log p \rfloor$ number of samples from each SEM for $c \in \{3, 4, ..., 13\}$.

## 5.2 COMPARISON AGAINST BASELINES

For experiments with a finite number of samples, we follow our generative process above. We generate 30 pairs of SEMs with $p \in \{10, 20, 50, 100\}$. We then generate $\lfloor e^c \log p \rfloor$ number of samples from each SEM for $c \in \{9, 10\}$. In Figure 3, we compare against the algorithms: PC (Spirtes et al. 2000) and GES (Meek 1997), both of which first learn each SEM *separately* and then output the difference of adjacency matrices as the difference DAG. For GES and PC, the undirected edges are oriented according to the true graphs, this way we provide a slight advantage to these methods for fair comparison. For PC we used Fisher tests and kernel-based tests for Gaussian and non-Gaussian noises. Finally, we also compare against the DCI-C method of Wang et al. (2018), which, as in our setting, also estimates the difference of SEMs. We note how traditional state-of-the-art methods (PC and GES) struggle to learn the difference DAG since

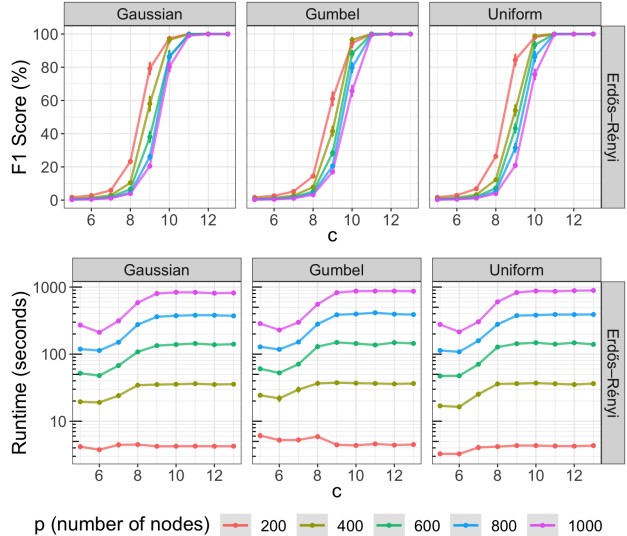

Figure 4: Performance vs sample size in high dimensions (up to 1000 nodes). See Section 5.1 for details on the data generation process. We note that our method is capable of learning the DDAG in around 15 minutes for $p = 1000$, and perfectly recover the causal changes as the number of samples increases.

each DAG independently is *dense*. The closest to our results is the DCI-C method, although, as seen in Figure 3, our algorithm performs better in both the F1-score and the runtime. We also compare the performance of our method on different sample sizes for graphs up to 100 nodes. As shown in Figure 2, our method learns the perfect difference DAG in all noise distributions for $c \geq 12$, within a few seconds.

## 5.3 EXPERIMENTS IN HIGH DIMENSIONS

We next present experiments in high dimensions (where the number of variables $p$ is large, up to 1000 nodes) to evaluate the performance of our algorithm. We do not compare against baselines in this setting as the baselines are not scalable to high dimensions, as can be seen from Figure 3, where the baselines take 1000s of seconds to run on graphs with 100 nodes, while our algorithm takes only a few seconds. Similar to the low-dimensional setting, the metrics are averaged over 30 runs. Here $p \in \{200, 400, 600, 800, 1000\}$. Our algorithm is able to perfectly recover the difference DAG, as shown in Figure 4, for $c \geq 12$.

## 6 CONCLUSION

We studied the problem of directly estimating the difference DAG of two linear SEMs. We presented novel conditions for the identifiability of causal shifts between linear SEMs leveraging the information encoded in the difference of precision matrices. By analyzing the strategy for removing terminal vertices, we showed the importance of minimal topological layering and its implications on the edge requirements for the difference DAG $\Delta$. Our findings not only provide a deeper understanding of the structural constraints necessary to recover the structure of $\Delta$, but also pave the way for the development of more efficient and accurate algorithms to learn the difference DAG between SEMs, even in high-dimensional settings.

## Acknowledgements

K. B. was supported by NSF under Grant #2127309 to the Computing Research Association for the CIFellows 2021 Project. We express our gratitude to our colleagues and reviewers for their valuable feedback and insightful comments, which greatly contributed to the improvement of this work. Additionally, we are grateful for the support of the University of Chicago Research Computing Center for assistance with the calculations carried out in this work.

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

# SUPPLEMENTARY MATERIAL
## Identifying Causal Changes Between Linear Structural Equation Models

## A  DETAILED PROOFS

### A.1  PROOF OF PROPOSITION 1

*Proof.* Consider a terminal vertex $i$ in $\Delta$. By definition, a terminal vertex in $\Delta$ implies that there are no outgoing edges from vertex $i$ to any other vertex in the difference graph. This means that for any $l \in \phi^{(1)}(i) \cup \phi^{(2)}(i)$, the difference in the connection strengths, $B_{l,i}^{(1)} - B_{l,i}^{(2)}$, must be zero, as there is no influence from vertex $i$ to vertex $l$ in the difference graph.

Therefore, for each $l \in [p]$, either $B_{l,i}^{(1)} = B_{l,i}^{(2)}$ or both are zero. Consequently, the product $(B_{l,i}^{(1)} + B_{l,i}^{(2)})(B_{l,i}^{(1)} - B_{l,i}^{(2)})$ becomes zero for all $l \in [p]$. As a result, every term in the sum of Equation 1 is zero, leading to $\Delta_{\Omega_{i,i}} = 0$. $\square$

### A.2  PROOF OF PROPOSITION 2

*Proof.* Assume $i$ is a terminal vertex in $G^{\cup}$. Since $G^{\cup} = G^{(1)} \cup G^{(2)}$, being a terminal vertex in $G^{\cup}$ means that vertex $i$ has no outgoing edges in both $G^{(1)}$ and $G^{(2)}$. Now, consider $\Delta$, which represent the differences in edges between $G^{(1)}$ and $G^{(2)}$. Hence $\Delta$ is a subgraph of $G^{\cup}$. Since $i$ is a terminal vertex in both $G^{(1)}$ and $G^{(2)}$, there can be no edges originating from $i$ that would be present in one graph and absent in the other. Therefore, in $\Delta$, vertex $i$ cannot have any outgoing edges, making it a terminal vertex in $\Delta$ as well. $\square$

### A.3  PROOF OF LEMMA 1

*Proof.* Let $\Delta_{\Omega_{i,i}} = 0$ for some $i$. From Assumption 1, we get $i$ is a terminal of $G^{\cup}$, i.e. $i$ is a terminal in both SEMs. From Proposition 2 from Ghoshal et al. Ghoshal & Honorio (2018), we know that the non-diagonal entries of the precision matrix $\Omega$ of an SEM $(B, D)$ are given by:

$$\Omega_{i,j} = -\frac{B_{i,j}}{\sigma_i^2} - \frac{B_{j,i}}{\sigma_j^2} + \sum_{l \in [p]} \frac{B_{l,i} B_{l,j}}{\sigma_l^2}.$$

So, if $i$ is a terminal in the SEM $(B, D)$ i.e. $B_{l,i} = 0, \forall l$, then $\Omega_{i,j} = -\frac{B_{i,j}}{\sigma_i^2}$.

In our case $i$ is a terminal in both SEMs, therefore $\Delta_{\Omega_{i,j}} = \Omega_{i,j}^{(1)} - \Omega_{i,j}^{(2)} = \frac{-B_{i,j}^{(1)} + B_{i,j}^{(2)}}{\sigma_i^2} = -\frac{\Delta_{B_{i,j}}}{\sigma_i^2}$ $\square$

### A.4  PROOF OF PROPOSITION 3

*Proof.* **Part 1: Iterative Removal and Topological Ordering.** Let us begin by establishing that the process of iteratively removing terminal vertices one-by-one from $\Delta$ is equivalent to removing them in reverse order of any topological ordering of $\Delta$.

Let $\theta$ represent an order of removing terminals of $\Delta$, i.e., $\forall i \in [p], \theta_i$ is the terminal removed from the remaining subgraph of $\Delta$ at $i$th step. This means that $\theta_i$ doesn't have any successor in the remaining subgraph of $\Delta$ at $i$th step, i.e., $\forall j > i$, $\Delta$ doesn't have an edge from $\theta_i$ to $\theta_j$. Hence making reverse of $\theta$ a topological order of $\Delta$.

Conversely, consider any topological ordering $\tau$ of $\Delta$. If we remove vertices in the reverse order of $\tau$, we always remove a terminal vertex of the remaining subgraph of $\Delta$ at each step, i.e., $\forall m \in [p], \tau_m$ is a terminal in $\Delta_{[m,\tau]}$. This is because in a topological ordering, all the successors of a vertex come after the vertex itself.

**Part 2: Converse of Proposition 2 and Topological Orderings.** Assume that the converse of Proposition 2 holds after every iterative removal of a terminal vertex from $\Delta$. The converse of Proposition 2 states that if a vertex is terminal in

$\Delta$, then it is also terminal in $G^{\cup}$. This implies that the removal sequence prescribed by some topological ordering of $\Delta$ also represents a valid topological ordering for $G^{\cup}$. Since the choice of the topological ordering of $\Delta$ was arbitrary, every topological ordering of $\Delta$ must be a valid topological ordering of $G^{\cup}$.

Conversely, suppose that every topological ordering of $\Delta$ is a topological ordering of $G^{\cup}$. And since $\Delta$ is a subgraph of $G^{\cup}$, every topological ordering of $G^{\cup}$ is also a topological ordering of $\Delta$, therefore $G^{\cup}$ and $\Delta$ have the same set of topological orderings. Then, the iterative removal of terminal vertices from $\Delta$ according to any of its topological orderings does not introduce a terminal vertex in $G^{\cup}$ that is not terminal in $\Delta$. This ensures the validity of the converse of Proposition 2 throughout the iterative removal process.

**Part 3: Transitive Edges and Topological Orderings.** Finally, we prove the claim regarding transitive edges. Let $G$ be a DAG and $H$ be a subgraph of $G$. The set of topological orderings of $H$ is the same as that of $G$ if and only if all edges of $G$ missing in $H$ are transitive edges of $G$.

Let an edge from $v$ to $u$ in $G$ is missing in $H$. If it is not a transitive edge of $G$, then absence of this edge in $H$ allows new topological ordering for $H$, not valid for $G$. One such ordering can be formed by first placing all the non-successors of $v$ in $G$, excluding $v$, in their topological order, followed by $u$, followed by $v$, last followed by all the successors of $v$ in their topological order. This is a valid topological ordering of $H$, but not for $G$ because $u$ comes before $v$.

Conversely, if all missing edges in $H$ are transitive in $G$, their removal does not create new topological orderings, as there are alternative paths preserving the precedence relations. Thus, every topological ordering of $G$ remains valid for $H$. □

## A.5 PROOF OF LEMMA 2

*Proof.* The converse of Proposition 2 implies that the set of terminals of $\Delta$ is same as the set of terminals $G^{\cup}$. The iterative process of removing terminals all-at-once can be described as:

- Initially, set of terminals of $\Delta$ = set of terminals of $G^{\cup}$. Let $L_0$ be the set of terminals of $\Delta$. Let $\Delta_{-L_0}$ be the DAG obtained after removing $L_0$ from $\Delta$. Similarly we define $G^{\cup}_{-L_0}$.

- Set of terminals of $\Delta_{-L_0}$ = set of terminals of $G^{\cup}_{-L_0}$. Let $L_1$ be the set of terminals of $\Delta_{-L_0}$. Let $\Delta_{-(L_0 \cup L_1)}$ be the DAG obtained after removing $L_1$ from $\Delta_{-L_0}$. Similarly we define $G^{\cup}_{-(L_0 \cup L_1)}$.
  ...

- Set of terminals of $\Delta_{-(\cup_{i=0}^{k-1} L_i)}$ = set of terminals of $G^{\cup}_{-(\cup_{i=0}^{k-1} L_i)}$. Let $L_k$ be the set of terminals of $\Delta_{-(\cup_{i=0}^{k-1} L_i)}$ and also equal to the set of all the vertices in $\Delta_{-(\cup_{i=0}^{k-1} L_i)}$. (Process stops!)

This iterative requirement of converse of Proposition 2 is equivalent to the set of level-wise terminals of $\Delta$ and $G^{\cup}$ being the same. Here the level of a vertex is $r$ if it was removed as part of the set $L_r$ as described in the process above. Level of a vertex in a DAG can be defined using the recursive process as shown above or as the maximum length of a path starting from the vertex in the graph. Hence the iterative assumption of the converse of Proposition 2 for simultaneous removal of terminal vertices can be stated as: levels of all vertices in $\Delta$ and $G^{\cup}$ are the same, which is equivalent to minimal topological layerieng of $\Delta$ being a valid topological layering of $G^{\cup}$. □

## A.6 PROOF OF THEOREM 1

*Proof.* Consider the following two pairs of SEMs over three nodes:

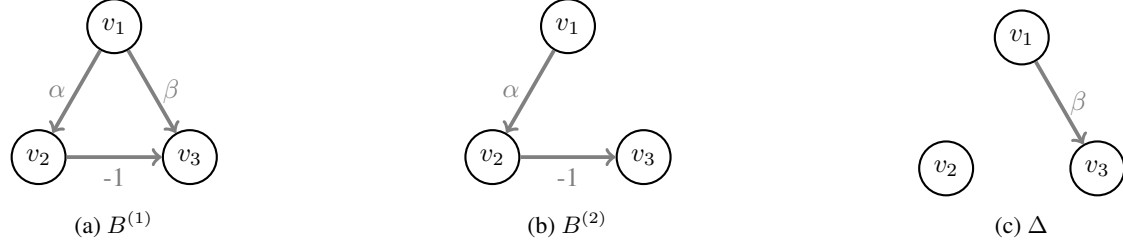

Figure 5: First pair of SEMs.

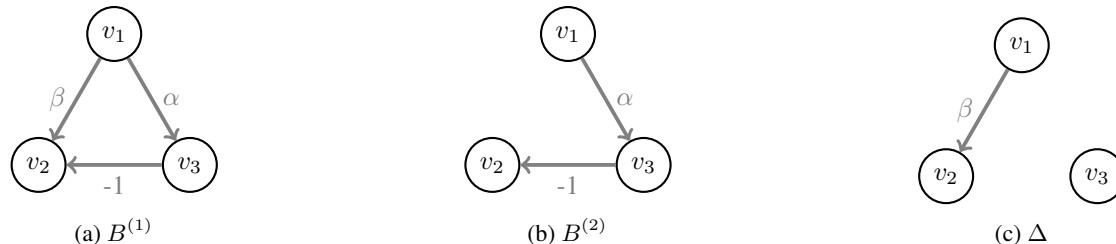

(a) $B^{(1)}$        (b) $B^{(2)}$        (c) $\Delta$

Figure 6: Second pair of SEMs.

Here $\alpha, \beta$ determine the edge weights of the SEMs and $\sigma$ is the variance of the exogenous noise variables. Then the difference precision matrix for both pairs of SEMs is

$$\frac{1}{\sigma^2} \begin{bmatrix} \beta^2 & -\beta & -\beta \\ -\beta & 0 & 0 \\ -\beta & 0 & 0 \end{bmatrix}$$

Both pairs of SEMs don't satisfy Assumption 2, while they do satisfy Assumption 3. We directly extend this to $p$ vertices, where $p$ let's say is a multiple of 3 and every 3 consecutive nodes can correspond to one of the two choices of pairs of SEMs. This gives us an exponentially large set of $2^{\frac{p}{3}}$ pairs of SEMs each having the same difference precision matrix as shown below.

$$\frac{1}{\sigma^2} \begin{bmatrix} \beta^2 & -\beta & -\beta & 0 & 0 & 0 & \cdots & 0 & 0 & 0 \\ -\beta & 0 & 0 & 0 & 0 & 0 & \cdots & 0 & 0 & 0 \\ -\beta & 0 & 0 & 0 & 0 & 0 & \cdots & 0 & 0 & 0 \\ 0 & 0 & 0 & \beta^2 & -\beta & -\beta & \cdots & 0 & 0 & 0 \\ 0 & 0 & 0 & -\beta & 0 & 0 & \cdots & 0 & 0 & 0 \\ 0 & 0 & 0 & -\beta & 0 & 0 & \cdots & 0 & 0 & 0 \\ \vdots & \vdots & \vdots & \vdots & \vdots & \vdots & \ddots & \vdots & \vdots & \vdots \\ 0 & 0 & 0 & 0 & 0 & 0 & \cdots & \beta^2 & -\beta & -\beta \\ 0 & 0 & 0 & 0 & 0 & 0 & \cdots & -\beta & 0 & 0 \\ 0 & 0 & 0 & 0 & 0 & 0 & \cdots & -\beta & 0 & 0 \end{bmatrix}$$

We can also make these SEMs connected by introducing an auxiliary vertex 0 which is connected to the topmost most vertex of all $\frac{p}{3}$ components. The difference precision matrix remains similar as before, only having one extra row and column of all zeros. $\qquad\square$

## A.7   PROOF OF THEOREM 2

*Proof.* Consider the following two pairs of SEMs over two nodes:

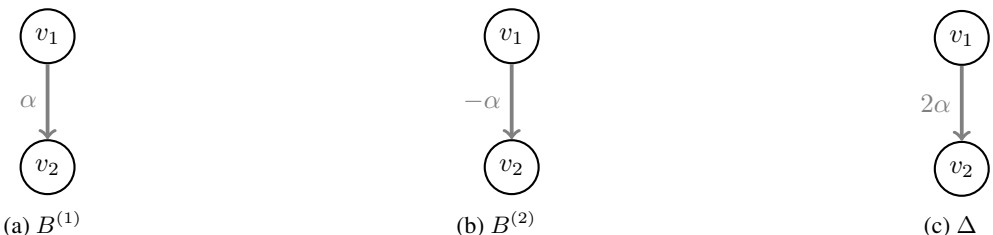

(a) $B^{(1)}$        (b) $B^{(2)}$        (c) $\Delta$

Figure 7: First pair of SEMs.

Here $\alpha$ determine the edge weights of the SEMs and $\sigma$ is the variance of the exogenous noise variables. Then the difference precision matrix for both pairs of SEMs is

$$\frac{1}{\sigma^2} \begin{bmatrix} 0 & -2\alpha \\ -2\alpha & 0 \end{bmatrix}$$

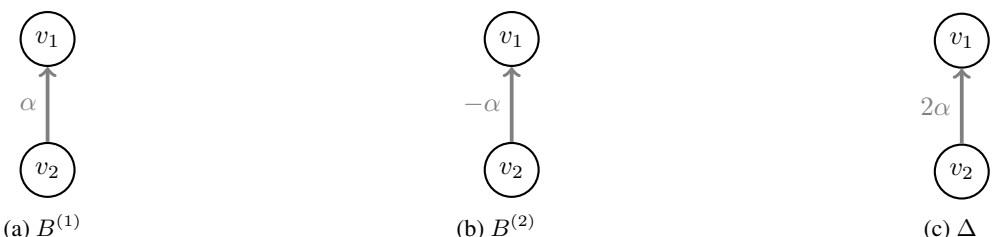

Figure 8: Second pair of SEMs.

Both pairs of SEMs don't satisfy Assumption 3, while they do satisfy Assumption 2. We directly extend this to $p$ vertices, where $p$ let's say is a multiple of 2 and every 2 consecutive nodes can correspond to one of the two choices of pair of SEMs. This gives us an exponentially large set of $2^{\frac{p}{2}}$ pairs of SEMs each having the same difference precision matrix as shown below.

$$\frac{1}{\sigma^2}\begin{bmatrix} 0 & -2\alpha & 0 & 0 & \cdots & 0 & 0 \\ -2\alpha & 0 & 0 & 0 & \cdots & 0 & 0 \\ 0 & 0 & 0 & -2\alpha & \cdots & 0 & 0 \\ 0 & 0 & -2\alpha & 0 & \cdots & 0 & 0 \\ \vdots & \vdots & \vdots & \vdots & \ddots & \vdots & \vdots \\ 0 & 0 & 0 & 0 & \cdots & 0 & -2\alpha \\ 0 & 0 & 0 & 0 & \cdots & -2\alpha & 0 \end{bmatrix}$$

We can also make these SEMs connected by introducing an auxiliary vertex 0 which is connected to the topmost most vertex of all $\frac{p}{2}$ components. The difference precision matrix remains similar as before, only having one extra row and column of all zeros. $\qquad\square$

## A.8 PROOF OF THEOREM 3

*Proof.* We prove Theorem 3 by induction on the number of variables in the system.

**Inductive Hypothesis:** Assume that Theorem 3 is true for all systems with $k$ or fewer variables, for some $k \geq 0$.

**Base Case:** For $k = 0$, the system has no variables, and the graphs are empty. Theorem 3 trivially holds in this case.

**Inductive Step:** Now, consider a system with $k + 1$ variables. In the first iteration of Algorithm 1, the set $S$ corresponds to the DN level-0 of $\Delta$. Note that $S$ is non-empty because the two SEMs share a topological ordering, therefore they have at least one common terminal, which will be in $S$. According to Assumption 3, for any vertex, its DN level is greater than or equal to its topological level. Hence, $S$ is the set of terminal vertices of $\Delta$, as the topological level of non-terminals is at least 1. From Assumption 2 and Lemma 2, these terminals are also terminals of $G^{\cup}$. Therefore, by Lemma 1, Algorithm 1 correctly identifies the incoming edges on this layer 0, as the corresponding non-zero entries in the row/column of the $\Delta_\Omega$. Since the variables in $S$ are terminals in both SEMs, removing them doesn't introduce any hidden confounders into the system. Thus, both SEMs remain causally sufficient Linear SEMs. Because we remove the variables in $S$ from the system all-at-once, Assumption 2 and Assumption 3 still hold in the new system. Therefore, we now have a smaller system with $k$ or fewer variables, under the same conditions. Hence, by induction hypothesis, Theorem 3 holds for this new system, i.e. Algorithm 1 will correctly identify the $\Delta$ of the remaining system, and we already identified the edges to $S$. Therefore, Algorithm 1 correctly learns the $\Delta$ for the system on $k + 1$ variables.

Therefore, by induction, Theorem 3 holds for any number of variables. $\qquad\square$