# OpenReview forum: "Identifying Causal Changes Between Linear Structural Equation Models"
_auai.org/UAI/2024/Conference — UAI 2024 poster_

### Official Review · Reviewer_y98e · 2024-03-01

**Q2-1 Originality-Novelty:** 2
**Q2-2 Correctness-Technical Quality:** 3
**Q2-5 Clarity Of Writing:** 3

**Q10 Ethical Concerns:**

No.

**Q1 Summary And Contributions:**

This paper aims to estimate the changes in directed acyclic graphs (DAGs) for linear models. The motivation is that directly learning the changes is significantly easier than the naive approach of learning the individual DAGs and taking their difference. By building on the previous work on precision matrix differences for linear SEMs, the paper considers the setting in which edge weights can change while statistics of the exogenous noise variables remain the same. The paper leverages the topological layering of the structural difference and proposes a simple efficient algorithm for estimating the difference DAG.

**Q2-3 Extent To Which Claims Are Supported By Evidence:**

2: Fair: the main claims are somewhat supported by evidence (but the experimental evaluation may be weak, or does not match entirely with the claims, important baselines may be missing, proofs contain important ideas but lack rigor, algorithmic details are only discussed superficially, references are imprecise, assumptions are not sufficiently motivated or explicated, etc.).

**Q2-4 Reproducibility:**

3: Good: key resources (e.g. proofs, code, data) are available and key details (e.g. proofs, experimental setup) are sufficiently well-described for competent researchers to confidently reproduce the main results.

**Q3 Main Strengths:**

- Direct estimation of the difference of DAGs is a relatively understudied problem (though there are recent works, )
- The paper provides useful observations and insights. For instance, explaining why in some cases identifying the difference of DAGs is possible even though the individual DAGs are nonidentifiable.
- Under certain assumptions, the proposed algorithm is very efficient, which is supported by the experiments.

**Q4 Main Weakness:**

**Novelty**: I think the main weakness of the work is limited novelty. To begin with, several important references are missing. Ghoshal et al. (2021) have a similar setting to Wang et al. (2018). To my knowledge, Ghoshal et al. is the first paper that repeatedly uses PDE (precision difference estimates) to learn the difference DAG. Another similarity, their algorithm also requires invariant noise variables.

Varici et al. (2021a) used similar ideas but studied a more general soft intervention setting, allowing noise variables to change upon intervention, and proposed CITE algorithm. Even though CITE primarily focuses on finding the intervention targets (i.e., terminal nodes of the difference DAG here), their paper also has results on estimating the structure (with results on causally insufficient systems given by Varici et al. (2022). For the same problem, Yang et al. (2023) is also a related reference).

In summary, I am having a hard time understanding the novelty of the results. Given the PDE-based results of Ghoshal et al. and Varici et al., I don’t think the contributions of this work are significant. Specifically, given the Assumptions of this work, algorithms proposed by Ghoshal and Varici can be modified without too much effort to yield the results in this paper. Furthermore, CITE algorithm of Varici et al. [2021a] is shown to be much faster than DCI, and uses the same ADMM-based PDE mechanism of Jiang et al. (2018) as this paper. Hence, the third point of the contributions paragraph seems rather insignificant.

Ghoshal et al. 2021. Direct Learning with Guarantees of the Difference DAG Between Structural Equation Models. arXiv 1906.12024, 2021.

Yang et al., 2023. Learning Unknown Intervention Targets in Structural Causal Models from Heterogeneous Data.  arXiv:2312.06091.

Varici et al., 2022. Intervention Target Estimation in the Presence of Latent Variables. UAI 2022

**Assumptions**:
- Assuming that the exogenous noise variables are invariant across two SEMs can be a strong assumption, yet not discussed properly. For instance, when considering an interventional setting which is a very likely case when considering two related SEMs, the exogenous noise variables are most likely to be affected.
- Assumption 1 is clear yet it is a quite strong assumption IMO. Converse of the Proposition 1 and Proposition 2 are explained after their definitions. The converse of Proposition 1 is milder, I can agree that certain assumptions may be necessary to rule out corner cases (perhaps the converse of Prop 1. can be related to a faithfulness condition?). However, the converse of Prop.2 can be easily violated.
- Given Assumption 1, Lemma 1 is a straightforward result of equation (1) (Prop.2 of Ghoshal and Honorio 2018b).

**Q5 Detailed Comments To The Authors:**

**Regarding Section 4.2** This can be a helpful observation, but I am not sure it’s important. My question is, once we identify all the terminal nodes, why do I care about removing them one by one instead of removing them all-at-once as Section 4.3 performs? For instance, for the statistical efficiency of the PDE procedure, reducing the number of total variables and the total tests is definitely desirable. Hence, I think it’s clear that removing the terminal nodes one-by-one is simply not desired, even without the theoretical roadblock. Am I missing something?

**Precision difference** To my knowledge, Jiang et al. only consider Gaussian RVs. So you can use it for experiments of course, but you should be careful for theoretical statements.

**Assumption 3 and do-interventions**: I think do-interventions would require more rigor when considering precision matrices, regarding the invertibility of the covariance matrices.

**Experiments**: In Section 5.2, the number of samples is a bit too large, e.g., $e^8 ~ 8103$. I’d expect to see lower numbers as well.

**Minor points**: Just some suggestions regarding notations and clarity:

- Repeatedly using the language “the converse of Proposition 1(or 2)” is not helping the reader.
- Pa and Ch are rather standard and easier notations than $\pi$ and $\phi$ to denote the parent and children sets.
- The meaning is apparent, but “terminal vertices” are not defined.
- Why does Figure 2 appear before the experiments section?
- Is minimal topological layering not defined in the literature? I can’t name a paper, but it felt too familiar.

**Q9 Complying With Reviewing Instructions:**

Yes

---

> ### Author Rebuttal · Authors · 2024-04-07
>
> Thank you for your thorough evaluation of our submission and for providing valuable feedback. Below, we address your main concerns:
>
> ### On Novelty
> * We discuss the paper by Ghoshal et al. (2021) in our rebuttal for all reviewers. Please see above.
> * Thanks for sharing the references by Varici et al. (2022) and Yang et al. (2023). We will add them in the revised version for additional context. However, **note that both works and also Varici et al. (2021) have different goals than ours**, namely, detecting intervention targets. Moreover, **Varici et al. (2021) does not estimate the DDAG (structural differences)** but the $\mathcal{I}$-MEC; and **although in their paper they allow for changes in the noise variances, they assume that the causal relations are invariant, i.e. there are no structural differences.** Please see the definition of soft intervention in Varici et al. (2021), eq.(4). We believe these distinctions are important and we hope you take them into consideration.
>
> ### On Assumptions
> * Regarding the assumption of invariant exogenous noise variables across SEMs, we note the following: (i) This assumption is unavoidable if our goal is to **identify (oriented) structural differences**, hence the same assumption is considered by the DCI algorithm, see Theorem 4.4 and Corollary 4.5 of Wang et al. (2018); (ii) This assumption is assumed in literature for learning causal graphs from multiple environments, see for instance Assumption 1 in [1].
>
> [1] Mameche et al. NeurIPS 2023. Learning Causal Models under Independent Changes.
>
> * Regarding the assumptions related to Propositions 1 and 2, we agree that the converse of Proposition 1 can indeed be seen as a form of a faithfulness condition. Regarding the converse of Proposition 2, its necessity is crucial for the identifiability of the exact structural differences (Delta) between the SEMs. **Without this assumption, as demonstrated in Theorem 1, identifiability would be limited to a super DAG of Delta.**
>
> ### On Section 4.2:
>
> To clarify, even with perfect estimation, the **all-at-once removal of terminal nodes is imperative for correctness**. Removing only a strict subset of terminal nodes (e.g., removing just one when multiple are present) can inadvertently introduce hidden confounders in the system defined over the remaining variables, where exact recovery of Delta is unlikely.
>
> ### On Precision Difference:
>
> Thanks for pointing this out! However, our identifiability theory treats the estimator as a black box, hence, we did not discuss it in detail. Interestingly, the estimator is fairly robust to misspecified noise distributions as shown in our experiments. We will add a short note on this.
>
> ### On Assumption 3 and do-interventions:
>
> The do-interventions we referred to only remove the causal relationships of the intervened variable with its parents, but do not hard set the value of that variable to a constant. This is also known as an “stochastic” hard intervention. We will make this more clear in the revision.
>
> ### On Experiments:
>
> We include experiments on sample complexity in Figure 2 (small graphs) and Figure 4 (large graphs) for our method. We did not include more values of $c$ in Figure 3 as we believe the figure would be harder to read. Nevertheless, we will include another figure similar to Figure 3 in the supplement for other values of $c$.
>
> We thank you for your other minor suggestions, we will make sure to address them in the revision. Please let us know if you have more concerns. We will truly appreciate it if you could re-evaluate our work based on our responses.

---

### Official Review · Reviewer_LGke · 2024-03-08

**Q2-1 Originality-Novelty:** 3
**Q2-2 Correctness-Technical Quality:** 3
**Q2-5 Clarity Of Writing:** 2

**Q1 Summary And Contributions:**

The paper describes a new method to identify from data the difference graph of two Linear Structural Equation Models. An implementation of the algorithm is proposed and the method is compared with existing approaches present in the literature.

**Q2-3 Extent To Which Claims Are Supported By Evidence:**

3: Good: the main claims are supported by convincing evidence (in the form of adequate experimental evaluation, proofs, (pseudo-)code, references, assumptions).

**Q2-4 Reproducibility:**

3: Good: key resources (e.g. proofs, code, data) are available and key details (e.g. proofs, experimental setup) are sufficiently well-described for competent researchers to confidently reproduce the main results.

**Q3 Main Strengths:**

The problem under study is definitely interesting to the community. The paper is solid, and the relationship with existing work is well described.

**Q4 Main Weakness:**

-The assumptions made are not very intuitive; more examples should be provided.
-The connection to Intervention detection should come earlier and be discussed more throughout the paper. From my understanding this is the only practical situations in which one can be sure that the graphical conditions are satisfied.
-In the introduction, the point that other methods are not well suited for a finite sample analysis is made, but no such analysis is presented for the proposed method.

**Q5 Detailed Comments To The Authors:**

Here I list a few minor comments.

- The abstract is too long
- Remark 1 is incorrect/imprecise? —> Cauchy distributions are allowed in LiNGAM models but not by your approach.
- Terminal vertices are not defined.
- Is assumption 1 satisfied for a generic choice of parameters?
- Missing ref. There is a new branch of the literature on causal discovery, aiming to produce recursive algorithms that are similar in spirit to the one proposed here, see, e.g., [Mokhtarian et al., 2021, JMLR].
- Why is the gaussianity assumption in sec 4.3 necessary?.
- “Note the minimal topological layering of a DAG is unique.” should go before Lemma 2 and an explanation is required.
- Can the method be used to detect intervened nodes?
- Can one interpret the output of the algorithm when (some of) the assumptions are violated?
- In the experiment section, the sample size should be mentioned earlier. Is the “c” in Fig. The same as the one in Fig 3? Why does the running time decrease when the sample size increases in the Low dimensional regime? And not in the High dimensional one?

**Q9 Complying With Reviewing Instructions:**

Yes

---

> ### Author Rebuttal · Authors · 2024-04-07
>
> Thank you for your positive evaluation of our work and for providing valuable feedback! We appreciate the opportunity to clarify and enhance the presentation of our work. Below, we address your comments:
>
> ### On more intuition on the assumptions
>
> Good point! Given the extra pages for the camera ready, we plan to move the examples in Figures 5 and 7 in the Appendix to the main text as we think they provide good intuition on the role of Assumptions 2 and 3.
>
> ### On connections to Intervention Detection
>
> Fair point, we will adjust our discussion on interventions for the revision. However, it is worth reminding that our goal in this work goes beyond detecting intervention targets, we also aim to identify the changes in causal influences (i.e., changes on the edge weights).
>
> Also, an additional practical scenario where our problem setting is applicable includes time-varying models, as discussed by Giannakis et al. (2018), where data samples are divided into possibly overlapping windows. For linear SEMs, Natali et al. (2021) describe the model as $X_t = B_t X_t + e_t$. Considering scenarios where the strength of causal relationships diminishes over time, such as the waning efficacy of a vaccine against disease resistance, demonstrates the relevance of our problem setting in practical environments.
>
> Giannakis, Georgios B., Yanning Shen, and Georgios Vasileios Karanikolas. "Topology identification and learning over graphs: Accounting for nonlinearities and dynamics." Proceedings of the IEEE 106.5 (2018): 787-807.
>
> Natali, Alberto, et al. "Online graph learning from time-varying structural equation models." 2021 55th Asilomar Conference on Signals, Systems, and Computers. IEEE, 2021.
>
> ### Finite Sample Analysis
>
> We did not delve into a detailed sample complexity analysis because the primary focus of our paper is on identifiability of the DDAG. However, it is worth noting that the sample complexity of our algorithm follows straightforwardly from the sample complexity of the difference of precision matrices estimator; since we use the estimator by Jiang et al. (2018), we can make use of their finite-sample rates, see for instance Theorem 1 in Jiang et al. (2018). We will include a short note on this in the revision.
>
> ### Minor Comments
>
> * **Re: Abstract.** We agree! We will make the abstract more concise in the revision, ensuring it remains representative of our paper's core contributions.
> * **Re: Remark 1.** Thanks for catching this! Your point is correct, we will make the remark more precise. Our model does not cover all LinGAM models but a large part of them.
> * **Re: terminal vertices.** We will define terminal vertices in the Preliminaries to clear any ambiguity.
> * **Re: Assumption 1.** Assumption 1 is not satisfied for a generic choice of parameters, meaning that there exists parameters of the SEMs where Assumption 1 might be violated. It might be helpful to think of this assumption as a milder form of faithfulness.
> * **Re: Mokhtarian et al., 2021.** Thanks for sharing the reference, we will include it in the revision.
> * **Re: Gaussianity assumption.** We are not sure if it was a typo on your side, but **we do not assume Gaussianity at all** for our theoretical results.
> * **Re: minimal topological layering.** We will add a short note on that.
> * **Re: detection of intervened nodes.** Yes, our algorithm can be used to detect intervened nodes, in this case, the nodes with incoming edges in the DDAG are intervened nodes.
> * **Re: violation of assumptions** Great question! Yes, for example, even if the structural assumption (Assumption 2)  is not satisfied, our algorithm can still output a super structure of the DDAG under some form of faithfulness assumption. We will add a note on that.
> * **Re: sample complexity experiments.** Yes, $c$ is the same across our set of experiments, it is used to define our number of samples as $n=e^c \log p$. Regarding the running time, the large errors in estimation (when the number of samples is small) led to more steps of our algorithm when compared to a more accurate estimation (when number of samples is larger).
>
> We thank you again for your comments, and we hope our answers make you feel more positive about our work. Please let us know if you have any additional questions!

---

### Official Review · Reviewer_duRt · 2024-03-12

**Q2-1 Originality-Novelty:** 3
**Q2-2 Correctness-Technical Quality:** 3
**Q2-5 Clarity Of Writing:** 3

**Q1 Summary And Contributions:**

This paper demonstrates that structural causal changes (in the form of a difference graph) are identifiable under some conditions, even when samples are obtained from two linear SCMs in cases where the noise is Gaussian with unequal noise variances (graphs representing a linear SCM with Gaussian noise and unequal noise variances are known to be non-identifiable.) Moreover, the authors compare two identification strategies: the first involves removing terminals one by one, while the second involves removing all terminals simultaneously, demonstrating that the latter is more efficient. Building on this second strategy, they propose a new algorithm for discovering the difference graph, which is validated theoretically and on simulated data.

**Q2-3 Extent To Which Claims Are Supported By Evidence:**

3: Good: the main claims are supported by convincing evidence (in the form of adequate experimental evaluation, proofs, (pseudo-)code, references, assumptions).

**Q2-4 Reproducibility:**

3: Good: key resources (e.g. proofs, code, data) are available and key details (e.g. proofs, experimental setup) are sufficiently well-described for competent researchers to confidently reproduce the main results.

**Q3 Main Strengths:**

* A novel identifiability result for detecting difference graphs.
* Novel algorithm for discovering difference graphs with theoretical guarantees.
* Related works are clearly discussed
* Experiments on simulated data demonstrate the performance of the presented algorithm
* Experiments conducted on simulated data showcase the efficacy (in terms of performance and computation time) of the proposed algorithm.

**Q4 Main Weakness:**

* The paper's presentation is somewhat confusing, particularly regarding the interrelation between Subsections 4.1, 4.3, and 4.4. Is Assumption 1 necessary in Subsection 4.3? (I think not, but this is not clear in the text)
* The algorithm was not tested on real data.
* Experimental results were not thoroughly  disscussed.

**Q5 Detailed Comments To The Authors:**

* I believe the title and abstract could be misleading, as one might interpret the paper as investigating causal changes at the level of the SEM (i.e., structural + parametric changes). To avoid this ambiguity, I suggest that the authors explicitly state from the outset that they focus solely on structural changes (and perhaps consider adding this clarification to the title). In this context, the example presented in Figure 1 is perplexing as it suggests a parametric difference rather than a structural difference. Am I missing something?
* There is minor ambiguity in the introduction's opening statements. The authors assert that a Structural Equation Model (SEM) effectively represents causal relations, but they also mention that the graph associated with the SEM is unknown. A reader who is not familiar with SEMs and graphs might mistakenly believe that the SEM itself is known while the graph is unknown. Clarification is needed in this regard.
* The immediate impact of the work introduced in this paper on the methods cited for root cause analysis may not be apparent. However, two recent methods introduced by Assaad et al. (2023) and Li et al. (2022) are likely to directly benefit from the contributions of this paper. For example, in Assaad et al. (2023) (where the SEM is also assumed to be linear), they estimate the difference in causal changes between normal and anomalous regimes based on a causal graph of the normal regime to detect root causes. I am confident that the work presented in this paper will address the limitations of these approaches, particularly concerning the non-identifiability of the causal graph of the normal regime. I encourage the authors to discuss these direct impacts to engage a broader community.

* In the related works section, I recommend clearly specifying which papers demonstrated identifiability for non-Gaussian noise (i.e., Shimizu et al., 2006) and which papers demonstrated identifiability for Gaussian noise with equal variances. This would enable readers unfamiliar with the subject to directly refer to these papers to understand the challenges as well as the significance of your paper.
* In Figure 1 : the numbers (i.e., coefficients) on the edges are not defined in the caption.
* Remark 1 : the "n" in "LiNGAM" should be capitalized.
* I recommend emphasizing the assumption that the two DAGs share the same causal ordering. For instance, formally state this assumption in a specific box for clarity (as in Assumption 1 and 2).
* I recommend stating somewhere in the main paper that proofs are provided in Supplementary Material.
* In Assumption 3 : missing « or » between « greater than » and « equal ».
* Always use \ref to refer to the assumptions or to the equations. In particular, in the paragraph just after theorem 2.
* Terminal vertices are not defined.
* The introduction of Section 4 (i.e. the first paragraph in Section 4) does not encompass all subsections within it.
* It would have been interesting to compare the two strategies experimentally.
* Are the considered assumptions testable?


References :
* Charles K. Assaad, Imad Ez-zejjari, Lei Zan. Root Cause Identification for Collective Anomalies in Time Series given an Acyclic Summary Causal Graph with Loops. AISTATS, 2023.
* Mingjie Li, Zeyan Li, Kanglin Yin, Xiaohui Nie, Wenchi Zhang, Kaixin Sui, Dan Pei. Causal Inference-Based Root Cause Analysis for Online Service Systems with Intervention Recognition. KDD, 2022.

**Q9 Complying With Reviewing Instructions:**

Yes

---

> ### Author Rebuttal · Authors · 2024-04-07
>
> Thank you for providing valuable feedback and for your positive assessment of our work! Below, we address your major comments:
>
> ### On the Paper's Presentation and Assumption 1
>
> Assumption 1 is indeed not necessary for the discussions in Subsection 4.3. This assumption, introduced in Subsection 4.1, serves as a unification of the converses of both Proposition 1 and Proposition 2. In subsequent subsections, including 4.3, we dissect this overarching assumption into its component parts relevant to the discussion at hand.
>
> ### On Testing with Real Data
>
> Given that our core contributions center around the identifiability of the DDAG, we focused on synthetic experiments where we can evaluate the correctness of our algorithm. That said, we will be including experiments on the gene-expression data from DCI (Wang et al. 2018) for the revised version.
>
> ### On Experimental Results Discussion
>
> We have made efforts to evaluate the correctness of our algorithm (via F1 score) w.r.t. sample sizes in Figure 2  (small/medium graphs) and Figure 4 (large graphs). Moreover, we compared against classical baselines in Figure 3. Please let us know if you would like us to expand on some specific points we might have overlooked.
>
> ### On Title and Abstract Clarification
>
> We clarify that **our algorithm detects parametric changes**. For example, let $B_{ij}^{(1)} = 0.4$ and $B_{ij}^{(2)} = 0.8$, then $[\Delta_B]_{ij} = 0.4$ and our algorithm detects the edge $(i,j)$ as part of the DDAG.
>
> ### On Ambiguity of SEMs in the introduction
>
> Good point! We will make sure to clarify that both the parameters and the graphical structure are unknown.
>
> ### On related work in Root Cause Analysis
>
> Thanks for sharing the references by Assaad et al. (2023) and Li et al. (2022), these are nice connections that we were not aware of! We will add a brief discussion on them in the revision.
>
> ### On Clarifying existing identifiability results
>
> Fair point. We will make sure to revise that. To be precise, the identifiability of Gaussian noises with equal variances were proven in [Peters & Bühlmann, 2014] and [Loh & Bühlmann, 2014]; while the identifiability of LinGAM is given in [Shimizu et al. 2006].
>
> ---
> Thank you for your remaining suggestions, we will make sure to include them for the revised version. We hope our responses will help clarify your comments and make you feel more positive about our work. Please let us know if you have any additional questions.

---

### Official Review · Reviewer_MieZ · 2024-03-21

**Q2-1 Originality-Novelty:** 1
**Q2-2 Correctness-Technical Quality:** 3
**Q2-5 Clarity Of Writing:** 4

**Q1 Summary And Contributions:**

The paper proposes identification criterions as well as algorithm for difference DAGs. Assuming two DAGs have same topological ordering, the algorithm can recover difference by carefully removing sinks in a layered fashion, even when the DAGs are themselves dense and non-identifiable, as long as their difference DAG is.

**Q2-3 Extent To Which Claims Are Supported By Evidence:**

4: Excellent: all claims are supported by very convincing evidence (in the form of comprehensive experimental evaluation, rigorous mathematical proofs, detailed (pseudo-)code, precise references, well-motivated and realistic assumptions) and the authors deliver what they promise.

**Q2-4 Reproducibility:**

2: Fair: key resources (e.g. proofs, code, data) are unavailable but key details (e.g. proof sketches, experimental setup) are sufficiently well-described for an expert to confidently reproduce the main results.

**Q3 Main Strengths:**

Paper is well organized and theory is presented clearly with good explanation. Simulation setting is also very clear.

**Q4 Main Weakness:**

This work reproduces basically the same theory and algorithm of a 2019 paper: Direct Learning with Guarantees of the Difference DAG Between Structural Equation Models by Asish Ghoshal, Kevin Bello and Jean Honorio. The 2019 paper is not discussed or compared this submission. Though not explicitly discussing the layering structure, theorem 1 and algorithm 1 from the 2019 paper is identical to theorem 3 and algorithm 1 here.

**Q5 Detailed Comments To The Authors:**

See above comments. I suggest extensive comparison with the 2019 paper and examine if the results are already known.

**Q9 Complying With Reviewing Instructions:**

Yes

---

> ### Author Rebuttal · Authors · 2024-04-07
>
> Thank you for your comments. We have written a rebuttal to all reviewers concerning the comparison against Ghoshal et al. (2021); please see above. We will truly appreciate it if you could re-evaluate our work based on our responses.

---

### Official Review · Reviewer_e3Sj · 2024-03-24

**Q2-1 Originality-Novelty:** 3
**Q2-2 Correctness-Technical Quality:** 2
**Q2-5 Clarity Of Writing:** 3

**Q1 Summary And Contributions:**

This paper discusses a class of linear SEMs with identifiable structural differences. The class of the models allows distribution-free noise distributions including both Gaussian or non-Gaussian, and has different noise variances across the variables in the individual SEMs. Furthermore, it proposes a polynomial time algorithm for identifying causal changes between linear SEMs. The experimental results heuristically confirm that the proposed method achieves competitive performance in both accuracy and run-time when recovering the structural differences between two linear SEMs compared to previous works in finite sample settings.

**Q2-3 Extent To Which Claims Are Supported By Evidence:**

3: Good: the main claims are supported by convincing evidence (in the form of adequate experimental evaluation, proofs, (pseudo-)code, references, assumptions).

**Q2-4 Reproducibility:**

2: Fair: key resources (e.g. proofs, code, data) are unavailable but key details (e.g. proof sketches, experimental setup) are sufficiently well-described for an expert to confidently reproduce the main results.

**Q3 Main Strengths:**

The theoretical findings of the paper are novel, introducing a new class of models with identifiable structural differences. Using the novel conditions related to the topological layering of the structural difference and identical error variances of two groups, this paper proposes a computationally efficient algorithm to identify the difference DAG. Foremost, it is one of few algorithms for learning DAG difference.

**Q4 Main Weakness:**

However, in my opinion, there are still some points that require further clarification: (1) Feasibility of the Assumptions 2 and 3, (2) Unclear of Delta_{Omega} estimator, (3) Unclear Numerical Experiments, (4) Clarity Of Writing. The detailed comments are in Q5.

**Q5 Detailed Comments To The Authors:**

Major Comments:

1.	Feasibility of the Assumptions 2 and 3

The rationale behind Assumptions 2 and 3 is effectively elucidated across Sections 4.1, 4.2, and 4.3. Additionally, their necessity for learning difference DAG structures is emphasized in Theorems 1 and 2. However, there appears to be a relatively limited discussion regarding the feasibility of these assumptions. It could be beneficial to include a comparative analysis between the proposed Assumptions 2 and 3 and well-known assumptions like faithfulness and sparsity of the difference graph. This addition could enhance comprehension and insight into these assumptions.

Assumptions 2 and 3 (iterative constraint of the converse of Proposition 2 and 1, respectively) lack convincing justification. Specifically, Assumption 2 requires Delta to be dense in some cases, as mentioned in the paper, but there is no illustration of cases where Assumption 2 does hold. Additionally, Assumption 3's requirement for Delta to be dense is not discussed at all.

2.	Section 4.2 and 4.3 introduce two different approaches based on the converse of Proposition 2 to identify the difference graph. However, I am not convinced with the method called “removing terminals one-by-one" described in Section 4.2. Could you provide clarification on whether this approach offers any advantages compared to the latter approach?

3.	Unclear of Delta_{Omega} estimator

Remark 3 highlights that the overall computational complexity of the algorithm is O(p \times T(m) + p^2) where T(m) represents the time complexity of each estimation of the difference precision matrix and m denotes the current size of the set V. Given that m inevitably depends on p, the role of T(m) becomes crucial in ensuring that the computational complexity O(p \times T(m) + p^2) remains polynomial concerning p. Hence, Section 4.6 introduces the alternating direction method of multipliers (ADMM), providing estimates of the difference precision matrix in a computationally efficient manner. Since the computational complexity of ADMM is O(p^3), employing the ADMM estimator results in the overall computational complexity of the proposed algorithm becoming polynomial concerning p.

However, for ADMM to achieve computationally efficient as a consistent estimator, it needs to adhere to several fundamental assumptions. ADMM handles the sparse differences between two Gaussian linear SEMs and relies on the irrepresentability assumption that is prevalent in the l1-regularized penalty optimization. Hence, it is crucial to clearly outline the conditions under which the algorithm achieves polynomial time complexity and engage in a discussion regarding the compatibility of these conditions with the Assumptions 2 and 3 proposed in this paper.
.
4.	Experiments

It would be beneficial to provide a more detailed and precise description of the simulation settings. For example,

-	The phrase ‘expected O(p) edges’ to be somewhat ambiguous. It would likely improve comprehension of the simulation to provide precise parameters used in generating ER and SF models, such as the exact number of expected edges.

-	There is the statement that ‘edge weights are determined in a manner consistent with the identifiability conditions of Assumption 3’. Further elaboration on this specific aspect is needed.

-	The paper mentions that all simulation results are averaged over 30 runs. To ensure more reliable results, increasing the number of simulations to at least 100 is recommended.

-	The PC and GES algorithms output estimated SEMs that are consistent up to the Markov equivalence class (MEC). Hence, it appears that additional steps may be necessary for a fair comparison. A more detailed explanation on this matter would provide clarity and enhance understanding. Moreover, as the noise distribution changes, it is conceivable that the CI tests, likelihood functions and other options employed in the PC and GES algorithms would need to be adjusted accordingly. It would be essential to specify the settings used in each simulation to ensure reproducibility. Additionally, Elaboration on how these algorithms can be applied to find a difference DAG is required. Finally, detailed explanations of each algorithm, such as the type of conditional independence test used for the PC algorithm, would be helpful.

Moreover, improving the interpretation of the following aspects of the simulation results could lead to a better discussion.

-	In Figure 2, the runtime seems to decrease as the sample size increases, which contrasts with established beliefs. It would be advantageous to offer an explanation for this phenomenon.

-	Section 5.2 demonstrates the superiority of the proposed algorithm compared to other state-of-the art methods. However, a detailed explanation for its outperformance is lacking. Clarification on why the DCI method struggles to capture the true DAG and why the proposed algorithm outperforms other algorithms would be beneficial.

-	It would be beneficial to include comparisons with other algorithms that address the structural difference between DAGs, such as chen et al. (2023) or Varici et al. (2021a), instead of solely focusing on algorithms that learn up to the MEC, such as PC or GES algorithms. Additionally, some distribution-free linear SEM recovery algorithms should be compared such as the US and LISTEN algorithms in the settings that all error variances are the same.

Chen, Tianyu, et al. "iSCAN: Identifying Causal Mechanism Shifts among Nonlinear Additive Noise Models." Advances in Neural Information Processing Systems 36 (2024).

Varici, Burak, et al. "Scalable intervention target estimation in linear models." Advances in Neural Information Processing Systems 34 (2021): 1494-1505.

5.	Real data analysis

In order to showcase the practicality of this new algorithm, it would be advisable to consider incorporating a section dedicated to real data analysis.

Minor Comments:

	Remark 2 highlights that Gaussian SEMs with unequal noise variances are recognized to be unidentifiable. However, numerous recent works investigate the identifiability of Gaussian SEMs with unequal noise variances, as evidenced by studies such as:

  -	Ghoshal, Asish, and Jean Honorio. "Learning linear structural equation models in polynomial time and sample complexity." International Conference on Artificial Intelligence and Statistics. PMLR, 2018.

  -	Park, Gunwoong, and Youngwhan Kim. "Identifiability of gaussian linear structural equation models with homogeneous and heterogeneous error variances." Journal of the Korean Statistical Society 49.1 (2020): 276-292.

	On page 5, (Minimal) Topological Layering and Topological Level are introduced to establish the iterative necessity of the converse of Proposition 2 while removing terminals all-at-once. There are numerous recent works.

  -	Gao, Ming, Yi Ding, and Bryon Aragam. "A polynomial-time algorithm for learning nonparametric causal graphs." Advances in Neural Information Processing Systems 33 (2020): 11599-11611.
  -	Zhou, Wei, et al. "Efficient learning of quadratic variance function directed acyclic graphs via topological layers." Journal of Computational and Graphical Statistics 31.4 (2022): 1269-1279.
  -	Park, Gunwoong. "Computationally Efficient Learning of Gaussian Linear Structural Equation Models with Equal Error Variances." Journal of Computational and Graphical Statistics 32.3 (2023): 1060-1073.

These works use the concept of topological layering of a DAG. They observe that the minimal topological layering of a DAG is unique and utilize the uniqueness to develop computationally efficient algorithm for learning DAGs. Incorporating references to pertinent scholarly works, notably those aforementioned, would serve to elucidate the concept of topological layering for the readership.

	The expression Delta_{Omega_{ii}} appears to be used interchangeably with the expression (Delta_{Omega})_{i,i}.

	According to my understanding, the occurrence of (j,i) \in \Delta or j \in S, which is the negation of the 8-th row of the algorithm, is improbable if the Assumptions 2 and 3 are satisfied. Hence, I am intrigued as to whether any apprehensions arise regarding the reliability of the results \Delta, even in instances where the algorithm proceeds without interruption despite failing to satisfy the conditions stipulated in the if statement.

	In Figure 3, the label of x-axis d needs to be fixed with p. Additionally, it appears that the term undirect in the caption needs to be adjusted to undirected.

	In the Proof of Proposition 3, the first v in the second sentence below the Part 3 needs to be fixed with u.

	Incorporating further discussions on the limitations of this research and future works related to it in the conclusion section would likely enhance the richness of this analysis.

	On page 3, it is unclear why there are edges X_1 - X_4 and X_3 - X_4 in Figure 1.

	On page 3, there should be a period at the definition of Delta_{Omega}.

	On page 3, there should be a period at Equation (1).

	On page 4, there should be a bracket when referring to Equation (1).

	On page 6, The notation k is being used twice: Topological Layering in Definition 2, and DN Level-k variables.

	On page 8, there is no description for d in Figure 3. I think it would be p.

	On page 13, supp(Delta) should be supp(Delta_B).

I largely appreciate the main idea of extending identifiability conditions for the difference between two linear SEMs, which is indeed a novel concept. However, the paper falls short in providing a detailed discussion or justification of assumptions. Additionally, some numerical experiments fail to adequately support the theoretical findings, and the settings for these experiments are ambiguously outlined. Moreover, the presence of numerous typos further suggests that this paper is not yet ready for acceptance. I believe this paper could be suitable for publication in top-tier proceedings or journals after certain modifications. If the authors could offer detailed experiment settings and thorough explanations for the simulation results, I would be inclined to increase my score. Furthermore, a "solid" justification of the identifiability conditions, both theoretically and practically, would merit a very strong acceptance.

**Q9 Complying With Reviewing Instructions:**

Yes

---

> ### Author Rebuttal · Authors · 2024-04-08
>
> Thank you for your thorough evaluation of our submission and for providing valuable feedback! Below, we address to your main concerns:
> ### On Feasibility of Asms 2 and 3
> Asm 3 indeed can be thought of as form of faithfulness condition. Additionally we want to clarify that the iterative constraint of the converse of Prop. 2 requires Delta to be dense in some cases only if we would remove terminals one-by-one. Since we remove terminals all-at-once, then in no case Asm 2 requires Delta to be dense. There is a discussion provided on this at the end of Section 4.3. We didn't discuss Asm 3’s requirements on sparsity as it does not depend on it. We only discussed it for Asm 2 to show the advantage of removing terminals all-at-once.
> ### On the One-by-One approach
> The only advantage of one-by-one approach over all-at-once approach would be if the structural requirements for one-by-one removal are met (e.g. dense DDAGs as described above), then in a small sample size case where the entries don’t go close enough to zero, removing variables one-by-one would be more stable. However, it is worth noting that when such conditions are not met **the all-at-once removal of terminal nodes is imperative for correctness**. Removing only a strict subset of terminal nodes (e.g., removing just one when multiple are present) can inadvertently introduce hidden confounders in the system defined over the remaining variables, where exact recovery of DDAG will fail.
>
> ### On $\Delta_{\Omega}$
> We emphasize that our main contributions are related to the identifiability of the DDAG, and not to propose a new estimator of difference of precision matrices. For our theory, the $\Delta_{\Omega}$ estimator is a black box. Thus, by using any estimator with guarantees, we implicitly borrow its conditions for correctness. Under the same reasoning, any polynomial time estimator will result in a polynomial time complexity for our algorithm. ADMM was used because it provided the best time complexity along with great empirical performance for different types of random graphs and noise distributions. However we also tested with other estimators in the literature such as Zhao et al. (2014) and results were very similar, but much slower to obtain.
>
> ### Experiments
> * We agree we were a bit sloppy on describing the graphs generations. The first DAG is sampled to have $O(p^{1.75})$ expected edges, i.e., for an ER graph, the edge probability is $O(1/p^{0.25})$. To generate the second DAG, we consider a valid topological ordering from the first DAG, then each edge is independently chosen to change with probability $O(1/p)$ (including edge additions/removals/changes), giving a total of $O(p)$ expected edges for the DDAG.
> * **Re: Asm 3.** We sampled the weights randomly as described in Sec 5.1, if these sampled weights fail to satisfy Asm 3, then we simply sample again. In our experiments, we found that the assumption was very often easily satisfied from the first randomly sampled weights.
> * **Re: num of simulations.** We have not found major changes when running 100 simulations, except for slightly smaller standard errors. We will update our figures.
> * **Re: PC and GES.** Thanks for this point! For GES and PC, the undirected edges are oriented according to the true graphs, this way we provide a slight advantage to these methods for fair comparison. For PC we used Fisher tests and kernel-based tests for Gaussian and non-Gaussian noises.
> * **Re: Fig 2.** When the sample size is small, the errors in estimation lead to more steps of our algorithm (by failing to remove all terminals). In contrast, when more accurate estimations are obtained, our algorithm is observed to perform less steps. This is intuitive since our setting considers sparse DDAGs.
> * **Re: on DCI.** We cannot offer a formal statement on why DCI underperforms. We strongly believe that it is due to their algorithm relying on (kernel-based) hypothesis tests, which are known to require lots of samples. We didn't compare against DCI for larger  sample sizes since it gets prohibitively slow due to the cubic time complexity on it.
> * **On other baselines.** We didn't compare to Varici et al. (2021a) because their algorithm is not suited for our setting. They aim to estimate intervention targets, and also the $\mathcal{I}$-MEC, but **their method does not output the DDAG**. Regarding Chen et al. (2023), we didn't compare to iSCAN because our setting does not satisfy their assumptions; for example, see their Asm A where they consider that the functionals are nonlinear on each component, which is not true for linear models. Regarding LiSTEN and US, we will add both baselines in the revision; briefly, the story is similar to that shown in Figure 2 of Ghoshal et al. (2021), both methods can recover the DDAG **when noises have equal variances but are very sample inefficient**, moreover, they fail to recover the DDAG for general non-identifiable models such as Gaussian noises with unequal noise variances.

---

### Meta-Review · Area_Chair_4jiu · 2024-04-21

This paper discusses a class of linear SEMs in which the structural differences graph is identifiable. Furthermore, they introduce a polynomial time method to identify the difference graph.

The paper clearly contains novel results. Both theoretical and experimental results are valuable and interesting.
However, there are two main concerns as it is also pointed out by the reviewers:
One is about the presentation and lack of discussions about the validity of the assumptions.
The main assumptions need more justifications as they are quite abstract in the presented form.
The presentation of the paper can be improved specially by adding more details about the experiments. This concern has been addressed in the rebuttal by the authors.
The second concern is about a closely related work by Asish Ghoshal, Kevin Bello and Jean Honorio, “Direct Learning with Guarantees of the Difference DAG Between Structural Equation Models” 2019 which has not been cited and discussed by the authors. Although, the authors have cited several work from Ghoshal et al. This is not a good scientific practice. Note that the authors in their rebuttal claim that the result of the aforementioned work is not correct by showing an example.